# The Combined Effects of Milk Intake and Physical Activity on Bone Mineral Density in Korean Adolescents

**DOI:** 10.3390/nu13030731

**Published:** 2021-02-25

**Authors:** Jae Hyun Lee, Ae Wha Ha, Woo Kyoung Kim, Sun Hyo Kim

**Affiliations:** 1Department of Sport Science, College of Natural Sciences, Chungnam National University, Daejeon City 34134, Korea; leejh1215@cnu.ac.kr; 2Department of Food Science and Nutrition, College of Natural Science, Dankook University, Chungcheongnam-do, Cheonan City 31116, Korea; aewhaha@dankook.ac.kr (A.W.H.); wkkim@dankook.ac.kr (W.K.K.); 3Department of Technology and Home Economics Education, Kongju National University, Chungcheongnam-do, Gongju City 32588, Korea

**Keywords:** bone mineral density, milk intake, physical activity, adolescence

## Abstract

The purpose of this study was to examine the combined effects of milk intake and physical activity on bone mineral density in adolescents. This study was conducted using data from the 2009–2011 Korea National Health and Nutrition Examination Survey (KNHANES), which provided measurements of bone mineral density (BMD) in addition to basic health-related data. This study included 1061 adolescents aged 13 to 18 years (557 males and 504 females) whose data on milk intake and participation time in moderate to vigorous physical activity were available. BMD was measured by dual-energy X-ray absorptiometry (DXA). Milk intake was assessed using the 24-h recall method, and the levels of physical activity were examined using a questionnaire. The physical activity questions of 2009–2011 KNHANES were based on the Korean version of the International Physical Activity Questionnaire (IPAQ) short form. The subjects were classified into four groups according to milk intake and physical activity level: no milk intake + low-level physical activity group (M_no_P_low_), no milk intake + high-level physical activity group (M_no_P_high_), milk intake + low-level physical activity group (M_yes_P_low_), and milk intake + high-level physical activity group (M_yes_P_high_). The results of partial correlation controlling for age, body mass index (BMI), and energy intake showed that the BMD variables were associated significantly with physical activity in both males and females. Among males, the M_no_P_low_ group had the lowest BMD in all BMD variables, showing a significant difference from the high-level physical activity groups (M_no_P_high_, M_yes_P_high_) by multiple logistic regression analysis. Among females, the M_yes_P_high_ group showed a significantly higher lumbar BMD value than the other groups. The M_no_P_low_ group had approximately 0.3 to 0.5 times lower odds ratio for median or higher BMD values, compared to M_yes_P_high_ group. These results show that milk intake and physical activity have a combined effect on BMD, and suggest that to achieve healthy bone growth, it is important to encourage both moderate to vigorous physical activity and milk intake during adolescence.

## 1. Introduction

Bones are major organs that determine the body’s physique and perform various functions, such as protection of internal organs, mineral storage, and blood cell formation. Bone ossification begins in the prenatal period and almost reaches the total peak bone mass by the end of teenage growth [1,2]. During puberty, the bone mineral accrual rate reaches a peak, and approximately one quarter of the total bone minerals of adults are accumulated within two years at this time [3]. In Koreans, the peak bone mass of the femoral neck and total hip is achieved around the age of 20, and the greatest increase in lumbar bone mineral density (BMD) occurs between 11–13 years of age in females and 12–14 years of age in males [4,5]. Hence, adolescence is a very important period of life for the formation of healthy bones. People who fail to achieve optimal peak bone mass and strength during childhood and adolescence have been reported to be more likely to develop osteoporosis later in life [6,7], and low BMD is associated with a higher risk of fractures even in healthy children and adolescents, just as it is a risk factor for fracture in adults with osteoporosis [8,9]. Therefore, to obtain the benefits of healthy bones for life, appropriate interventions are required to help children and adolescents build healthy and strong bones during the growth period.

Peak bone mass, which means the maximum accumulation of bone mineral content, is determined by genetic and environmental factors. Environmental factors include physical activity, sedentary lifestyle, and dietary factors such as milk intake [10,11,12,13,14]. The consumption of milk and dairy products helps maximize the bone mineral content during puberty, which is the second period of the growth spurt [15,16]. Milk has a high calcium content, and calcium in milk has high digestibility and bioavailability [17]. This is because milk contains lactose, vitamin D, and peptides promoting calcium absorption, which help the body to absorb calcium, and contains calcium and phosphorus in an appropriate ratio that increases the rate of calcium absorption [18]. The consumption of milk and dairy products during the growth period can be a good source of calcium as well as energy, macronutrients, and micronutrients important for the growth and development of children and adolescents [13,14,17,18,19,20]. A four-year follow-up study of 19,991 children in eight European countries reported that the consumption of milk and dairy products (yogurt and cheese) as snacks was associated with better diet quality [21]. Therefore, the daily consumption of milk and dairy products for children and adolescents can be a good strategy for maintaining a balanced diet during the growth period. 

Mechanical stimulation is an important determinant of bone growth and formation. Exercises that provide physical and physiological stimulation improve muscular strength, cartilage preservation, and bone remodeling [22,23], and they have a positive effect on increasing BMD [24,25]. Most studies on the effects of weight-bearing exercises on the accumulation of bone mineral content during childhood and adolescence reported that such exercises have positive effects, and this phenomenon is particularly pronounced in early puberty [26]. The performance of the activities of high intensity or impact and participation in sports activities have also found to have a positive effect on the BMD or cortical bone size [25,27,28]. 

As described above, various studies have been conducted on the effects of milk intake or physical activity alone on BMD during the growth period. Limited studies suggested an important interaction between physical activity and the intake of dietary calcium, not milk intake, to increase bone mass. When physical activity and calcium intake were combined, bone density formation was greater than either physical activity or calcium intake alone [29,30,31]. In addition, those studies have been conducted in preschool or school children. Therefore, this study aims to evaluate the combined effects of milk intake and physical activity on BMD during adolescence. We hypothesized that adolescents who had a high level of physical activity and consumed milk would have higher BMD than those who had a low level of physical activity and did not consume milk. 

## 2. Methods

### 2.1. Data Collection

This study examined the relationship of BMD with milk intake and physical activity using 2009 to 2011 data from the fourth (2007–2009) and fifth (2010–2012) Korea National Health and Nutrition Examination Survey (KNHANES). The KNHANES survey began in 1998 and has been conducted annually, with BMD measurements conducted from July 2008 to May 2011. Data of 1731 people aged 13–18 (1198 males and 812 females) who underwent BMD measurements using dual energy X-ray absorptiometry (DXA) were collected. Subjects with missing data regarding milk intake or physical activity and those whose data showing extreme outliers were excluded. Ultimately, the data of 1061 people (557 males and 504 females) were included in the final analysis.

This study used the data from the KNHANES approved by the Institutional Review Board of the Korea Centers for Disease Control and Prevention (2009-01CON-03-2C, 2010-02CON-21-C, 2011-02CON-06-C), which was conducted after receiving an exempt determination from the Institutional Review Board of Kongju National University (KNU_IRB_2020-65). 

### 2.2. Milk Intake

The analysis of milk intake was conducted using data from a dietary intake survey by the 24-h recall method among the raw data sets of the KNHANES. According to the food-group classification standard codes presented in the guidelines on the use of the KNHANES data, the food name of ‘milk’ among the secondary food names was first classified. The type of milk consumed was then examined using the primary food names, and the participant was classified as a person consuming milk when the type of milk consumed was white milk.

### 2.3. Physical Activity

The level of physical activity was calculated by the time of moderate or vigorous physical activity performed per week (number of days per week (days/week) × activity time (minutes/day)). The questions on moderate and vigorous physical activity in KNHANES were as follows:Questions on moderate physical activity:
○On how many days in the past week did you perform moderate physical activity that made you feel slightly more tired than usual, or during which you felt a little short of breath for at least 10 min? ○On the days when you performed moderate physical activities, how many minutes per day did you usually perform them? 

Examples of moderate physical activities: vocational and physical activities, such as slow swimming, doubles tennis, volleyball, badminton, table tennis, moving, or carrying light items.

Questions on vigorous physical activity:
○On how many days in the past week did you perform vigorous physical activity that made you feel much more exhausted than usual, or during which you felt very short of breath? ○On the days when you performed vigorous physical activities, how many minutes per day did you usually perform them? 

Examples of vigorous activities: vocational and physical activities, such as jogging or running, mountain climbing, fast cycling, fast swimming, soccer, basketball, jumping rope, squash, singles tennis, moving or carrying heavy objects.

In this study, based on the guidelines on physical activity presented by the Ministry of Health and Welfare for calculating weekly physical activity time, it was assumed that one minute of vigorous physical activity is equal to two minutes of moderate physical activity [32]. Using this guideline, the total physical activity time was calculated by converting vigorous physical activity time to moderate physical activity time. The physical activity questions of the 2009–2011 KNHANES were based on the Korean version of the International Physical Activity Questionnaire (IPAQ) short form. 

### 2.4. Subject Grouping

The subjects were divided into the milk intake group (M_yes_ group: milk intake >0 g/day) and the no milk intake group (M_no_ group: milk intake = 0 g/day). For physical activity grouping, the median of the weekly participation time of moderate-to-vigorous physical activity was calculated by converting vigorous physical activity times to moderate physical activity time. Subjects with a value below the median were classified as the low-level physical activity (P_low_) group. Those with a value equivalent or higher than median were classified as the high-level physical activity (P_high_) group. Groups can also be classified according to the satisfaction of the physical activity guidelines of 60 min of moderate-to-vigorous activities every day. However, only 5.1% of men and 1.9% of women actually meet these criteria (420 min per week), making it impossible to compare the groups using statistical analysis. Therefore, in this study, groups were classified using the median of converted physical activity time per week. The physical activity questions of the 2009–2011 KNHANES were based on the Korean version of the International Physical Activity Questionnaire (IPAQ) short form. 

By combining these two classifications, the subjects were finally classified into four groups according to milk intake and physical activity level: no milk intake + low-level physical activity group (M_no_P_low_), no milk intake + high-level physical activity group (M_no_P_high_), milk intake + low-level physical activity group (M_yes_P_low_), and milk intake + high-level physical activity group (M_yes_P_high_). 

### 2.5. Bone Mineral Density

BMD was measured using dual-energy X-ray absorptiometry (DXA; DISCOVERY-W fan-beam densitometer Hologic Inc., Bedford, MA, USA) and each subject’s whole body, lumbar spine, and femur were scanned. When measuring the lumbar spine, a lumbar positioner was used to reduce spinal lordosis, and the lumbar spine was positioned straight so as to be in line with the vertical central axis of the image. The image included the mid-section of T12 and L5, and to determine whether the lumbar spine was correctly positioned, it was checked whether the 12th rib and iliac crest were visible in the image, and whether the intervertebral disc of L4–L5 passed in line with the iliac crest. When measuring the femur, the angle of the leg was adjusted so that the femoral shaft was positioned straight in line with the vertical central axis of the image. When measuring DXA, it was checked if there were any artifacts such as coins or keys, buttons, wires, jewelry, or metal objects in the pocket. Among the various DXA measurement indices, total body, femur, femur neck, and lumbar spine (L1–4) BMD were analyzed statistically, and total body BMD was calculated using the BMD values of the whole body except for head BMD.

### 2.6. Statistical Analysis

The data of the KNHANES were collected not by simple random sampling but by stratified multistage probability sampling. Hence, the weight, strata (KSTRATA), and cluster (primary sampling unit, PSU) were included in the analysis. The sociodemographic characteristics of the subjects were expressed as frequency and percentage, and differences in distribution between the groups were compared using PROC SURVEYFREQ (chi-squared test). For the continuous variables, descriptive statistical analysis was performed to calculate the mean and standard error. Partial correlation analysis was performed to identify the relationship of BMD with physical activity and milk intake while controlling for age, body mass index (BMI), and energy intake. The differences in explanatory variables between the four groups (M_no_P_low_, M_no_P_high_, M_yes_P_low_, and M_yes_P_high_ groups) were analyzed by PROC SURVEYREG analysis after adjusting for age, BMI, and energy intake. For a post-hoc test of the differences among the groups, the *p*-values were assessed using a Bonferroni test considering the design effect of complex sampling design. The PROC SURVEYLOGISTIC analysis was performed (after adjusting for age, BMI, and energy intake) to calculate the risk ratio of each BMD index of the three groups compared to the reference group (the M_yes_P_high_ group). The analysis results were expressed as an odds ratio (OR) and 95% confidence interval (CI).

All statistical analyses were conducted using SAS version 9.4 (Statistical Analysis System, SAS Institute, Cary, NC, USA), and *p* values <0.05 were considered significant.

## 3. Results

Table 1 lists the sociodemographic characteristics of the subjects. Significant differences in school year and gender were observed among the four groups. Of the 1061 subjects, high-school students (57.0%) comprised a larger proportion than middle-school students (43.0%), and the difference in the percentage between middle school and high school was the largest in the M_no_P_low_ group. The subjects consisted of 557 males (52.5%) and 504 females (47.5%), and the difference in the percentage between males and females was the largest in the M_yes_P_high_ group (68.0% in males vs. 32.0% in females). Therefore, the analysis was conducted separately for males and females, and data analysis was conducted by controlling for age. There were no significant differences in the distribution of income levels or residential areas.

Regarding the distribution of daily milk intake among subjects, the milk intake ranged from 0 to 1484 mL/day among males and from 0 to 848 mL/day among females. Approximately 55.4% of males and 62.6% of females did not consume milk, and in both males and females, the proportion of people drinking 200–400 mL/day was highest, accounting for 24.2% and 20.9%, respectively. According to the dietary reference intakes for Koreans (KDRIs), it is recommended that adolescents drink two glasses (400 mL) of milk a day [33], and the percentage of adolescents consuming the recommended amount or more of milk was 14.7% in males and 8.1% in females; females tended to drink less milk than males (Figure 1).

The converted time of physical activity ranged from 0 to 780 min/week among males and 0 to 600 min/week among females. The weekly participation time of moderate to vigorous physical activity except for walking was 0 min in 31.2% of males and 49.7% of females. For both males and females, the proportion of adolescents showing a converted physical activity time of 60–120 min per week was highest, accounting for 14.6% and 16.3%, respectively. The proportion of those participating in physical activity for 300 min or more per week was 12.9% in males and 4.9% in females. Hence, the level of participation in physical activity was significantly lower among females than among males (Figure 2).

Table 2 lists the milk intake and physical activity time of each group. Because the subjects were classified according to whether they consumed milk or not, the daily milk intake of the no milk intake groups (M_no_P_low_, M_no_P_high_) was 0 mL. In the milk intake groups, the milk intake levels for males in the M_yes_P_low_ and M_yes_P_high_ groups were 360.1 mL and 349.0 mL, respectively. For females, the milk intake levels in the M_yes_P_low_ and M_yes_P_high_ were 280.8 mL and 278.8 mL, respectively. There was a large difference in the physical activity time between the high-level physical activity groups (M_no_P_high_, M_yes_P_high_) and low-level physical activity groups (M_no_P_low_, M_yes_P_low_). For groups with high physical activity, weekly physical activity time among males was 227.3 min for M_no_P_high_ and 230 min for M_yes_P_low_. Among females, the physical activity time in the M_no_P_high_ and M_yes_P_high_ groups was 130.7 min and 175.9 min per week, respectively. The weekly physical activity time among males was 11.8 ± 1.9 min for M_no_P_low_ and 26.3 ± 3.7 min for M_yes_P_low_.

Table 3 lists the general characteristics of each of the four groups classified according to milk intake and the level of physical activity. In both males and females, the mean age was highest in the M_no_P_low_ group and lowest in the M_yes_P_high_. For BMI, there was a significant difference only in females, showing that the M_no_P_low_ and M_yes_P_low_ groups with low levels of physical activity had a significantly lower mean BMI than the groups with high levels of physical activity. However, the mean BMI of these four groups were not largely different from 20.9 kg/m ^2^, the median BMI (50th percentile) of 15.4-year-old boys in the 2017 Korean National Growth Charts for Children and Adolescents published by the Ministry of Health and Welfare [34]. For reference, the BMI corresponding to overweight (from the 85th percentile to less than the 95th percentile) for a 15.4-year-old Korean girl is 23.7~25.5 kg/m ^2^, and the BMI corresponding to obesity (95th percentile or more) is 25. 5 kg/m ^2^ or more [34].

There was no difference in body fat (%) between groups in both male and females. Regarding the percentage of body fat in each group, the lowest and highest mean values were 20.0% and 22.2% among males and 31.8% and 33.6% among females. For reference, the mean body fat percentages of the male groups correspond to the 50–75th percentile of the percent body fat of Korean male adolescents, and the mean body fat percentages of female groups correspond to the 25–75th percentile of Korean female adolescents [35].

In order to identify the association of physical activity and milk intake with BMD, a partial correlation analysis for each gender group was conducted while controlling for age, BMI, and energy intake (Table 4). The results of this analysis showed that milk intake had no significant correlation with BMD. On the other hand, physical activity was found to have a weak but significant correlation with total body, femur, femur neck, and lumbar BMD.

Table 5 lists the results of comparative analysis of BMD among the four groups. Among males, there was a significant difference among the groups in all BMD variables, and the M_no_P_low_ group, the group of adolescents who did not consume milk and had a low level of physical activity, had a significantly lower BMD than the M_no_P_high_ and M_yes_P_high_ groups, which had a high level of physical activity. The BMD values of the M_no_P_low_ group were lower than the median BMD value among 15-year-old Korean boys and higher than the 10th percentile [4]. In the case of females, there was a significant difference among the groups only in lumbar BMD. The M_yes_P_high_ group, the group of females who consumed milk and had a high level of physical activity, showed a significantly higher lumbar BMD value of 0.931 (g/cm ^2^) than the other groups (M_no_P_low_: 0.902, M_no_P_high_: 0.900, M_yes_P_low_: 0.898). For reference, the median lumbar BMD value among 15-year-old Korean girls was 0.875 g/cm ^2^ [4].

Table 6 lists the odds ratio and confidence interval (CI) for the 50th or higher percentile of the BMD value in each BMD variable for each group compared to the M_yes_P_high_ group. Among males, the M_no_P_low_ group had significantly lower odds ratio for the 50th percentile or higher of the BMD value than the M_yes_P_high_ group in all BMD variables. More specifically, the M_no_P_low_ group was 0.317 times less likely to have the 50th or higher percentile of total body BMD value than the M_yes_P_high_ group. For femur, femur neck, and lumbar BMD, the M_no_P_low_ group had 0.289, 0.512, and 0.493 times lower odds ratio for the 50th or higher percentile of the BMD compared to the M_yes_P_high_ group. In other words, the ratio of individuals with a median or higher BMD was significantly lower among the males who did not drink milk and had a low level of physical activity than the males who consumed milk and had a high level of physical activity. Among the females, the M_no_P_low_ group and M_yes_P_low_ group were 0.433 and 0.434 times less likely, respectively, to have the 50th or higher percentile of lumbar BMD than the M_yes_P_high_ group.

## 4. Discussion

Adolescence is a very important period for lifelong bone health. Several studies have reported that the factors that positively affect the increase in bone mineral content and density have greater effects during this period than in adulthood, and that the effects of such factors continue into adulthood [2,3,15,16,36,37].

The two methods for building strong bones or improving bone strength are ingesting sufficient nutrients related to the bone matrix or bone metabolism and applying appropriate mechanical stimulation to the bones. Typically, when the former method is used, people consume milk, which has a high calcium content and high digestibility and bioavailability of calcium. Weight-bearing physical activities are performed when the latter method is used. Consequently, the study was designed to examine the combined effects of milk intake and physical activity on BMD. 

In a partial correlation analysis controlling for age, BMI, and energy intake, physical activity had a significant positive correlation with total, femur, femur neck, and lumbar BMD in both males and females. Physical activity has beneficial effects on bone health in all age groups, including adolescents. In particular, bone mineral content is higher among children and adolescents participating in activities involving the exertion of high impact force than among those who participate in non-weight bearing exercises, such as swimming, or in low-impact activities, such as walking [38,39]. Therefore, activities involving high ground-reaction forces, such as jumping, skipping, and running, are recommended as exercises for strengthening the bones during the growth period [40,41]. A cross-sectional analysis of the relationship between physical activity and hip BMD in 724 adolescents found that high impact (>4.2 g) activities, such as jumping and running (speeds>10 km/h), were associated with hip BMD, but moderate impact activity, such as jogging, had little effect [25]. However, the physical activity variable analyzed in this study was the time of participation in moderate to vigorous physical activities. The physical activities with moderate intensity examined in this study included sports, such as slow swimming, doubles tennis, badminton, and table tennis. Walking was excluded from the analysis because it was examined separately with a different format. Given these facts, it seems that the low correlation between physical activity and BMD might be related to the type and intensity of physical activity analyzed in this study. Nevertheless, physical activity was consistently related to the BMD variable in both males and females.

On the other hand, milk intake and BMD had no significant correlation, which is inconsistent with previous studies reporting a quantitative relationship between the intake of milk and dairy products and bone mineralization. Several studies on the relationship between the intake of calcium, vitamin D, and dairy products and bone frailty during growth have reported conflicting or inconsistent results [42]. In this study, calcium and vitamin D intakes were not included as control variables. The reason is that it was confirmed that the calcium intake of Korean adolescents was very low, and milk was the major source of calcium. Besides, when the bone mineral density variable was analyzed using calcium as a parameter, the same result was obtained as from using milk intake. Also, vitamin D intake was excluded from the control variable as there was no significant difference between groups. There might be a threshold in the expression of the effect of calcium intake. Eating above a certain level of calcium does not affect the bone mass significantly but eating less than this can lead to an inadequate balance [1]. In addition, the effect of nutritional intake may vary depending on the nutritional status of the subjects. In cases where the intake of minerals or high-quality protein may be insufficient, the subjects may show a distinct increase in bone growth after the supply of dairy products [1,43]. On the other hand, Ren et al. reported that children with a good nutritional status did not show a clear positive correlation between major bone nutrients and bone outcomes compared to children with nutritional deficiencies [44]. In considering the results of this study that concern the relationship between milk intake and BMD, it is also necessary to consider that the overall milk intake level of the subjects was low, with an average daily milk intake of less than one glass (200 mL), and that 58.8% of subjects did not drink any milk. Therefore, additional studies will be needed to investigate the relationship between milk intake and BMD considering the distribution of milk intake and the basic nutritional status of subjects.

No linear relationship was observed between milk intake and BMD, but physical activity and milk intake had a statistically significant combined effect on BMD. Among males, the M_no_P_low_ group had the lowest BMD in all BMD variables, showing the statistical difference from the groups with a high level of physical activity, the M_no_P_high_ group and the M_yes_P_high_ group. Among females, the M_yes_P_high_ group had a significantly higher lumbar BMD than the other groups.

In particular, an analysis of the odds ratios of male subjects showed that those who did not consume milk and had a low level of physical activity (M_no_P_low_) were significantly less likely to have a high BMD than those who consumed milk and had a high level of physical activity (M_yes_P_high_). Specifically, the M_no_P_low_ group was approximately 0.5 times less likely to have a high femur neck BMD and lumbar BMD and was approximately 0.3 times less likely to have high BMD for the total body and the femur than the M_yes_P_high_ group. 

These results suggest that milk intake and physical activity have combined effects in strengthening bones. Branca et al. (2001) reported that bone anabolism could be increased by weight-bearing exercise during adolescence, and adequate calcium intake is necessary for exercises to have a bone stimulating effect [36]. In a review study on the interactions between physical activity and nutrients in children and adolescents, Julián-Almárcegui (2015) reported that the combined effects of exercise and calcium intake were greater than the effects of exercise or calcium intake alone, and physical activity required calcium intake to have a positive effect on bones [45]. According to a clinical report of the American Academy of Pediatrics, routine calcium supplementation is not required for healthy children and adolescents for bone health, and it is necessary to increase the supply of calcium through dietary intake to meet daily recommended levels [46]. Therefore, drinking milk, a major source of calcium, combined with moderate to vigorous physical activity that provides mechanical stimulation to the bones during growth, is considered an effective strategy to maximize bone growth potential.

In females, the effect of milk intake and physical activity was found only in lumbar BMD. The positive effect of physical activity or physical activity combined with nutrients on BMD was relatively insignificant in females, because the overall physical activity of females was low. The converted weekly moderate physical activity time was only 70.4 ± 47.9 min for females, compared to 120.7 ± 7.9 min for males. Probably due to the fact that a mechanical load has an impact on the bones in a region-specific and tissue-specific manner [44,47,48], there was no difference in the BMD of the femur and femur neck among groups, which are the areas where the mechanical impact is applied more directly during physical activity. In addition, the increase in the total body BMD and leg BMD slows down in females after the age of 13 years, whereas the lumbar BMD shows a relatively continuous increase during adolescence [4]. Therefore, the lumbar BMD of females seemed to reflect the effects of lifestyle more sensitively during adolescence. 

The results of this study showed that a combination of moderate to vigorous physical activity with milk intake during adolescence, which is a very important period for laying the foundation for lifelong bone health, is an effective strategy for maximizing the growth potential of BMD. Nevertheless, there is a need to consider the following limitations when interpreting and applying the findings of this study. First, this study was a cross-sectional survey study. A longitudinal study will be needed to elucidate and verify the causal relationships among physical activity, milk intake, and BMD with respect to the combined effects of the two factors on BMD. Second, the level of physical activity was assessed based on the participation time of moderate to vigorous physical activity, but the time spent walking was not included. The intensity of walking can vary from low to moderate. Although there was a separate questionnaire item on walking, it was excluded from the analysis because it did not quantify the intensity of walking. Third, the KNHANES used in this study does not investigate type of physical activity. Therefore, non-weight bearing physical activity participation time, such as slow swimming, included as an example of moderate intensity activity, could not be considered separately to be analyzed, possibly reducing the correlation between physical activity and BMD. Fourth, in this study, the level of physical activity was examined through a questionnaire survey. Hence, this study has inherent limitations regarding the objectivity and reliability of the self-report measures of physical activity, compared to objective, direct measures of physical activity. Fifth, when carrying out subject grouping, in terms of physical activity, the subjects were divided into high- and low-level physical activity groups. In the case of milk intake, however, because a considerable proportion of people did not consume milk, the subjects were classified into two milk groups: those who did not drink milk at all or those who did. In interpreting the results, it will be necessary to consider these differences in the criteria for evaluating impact of milk intake and physical activity. Sixth, due to dietary variation within the individual, there is a limit to grasp accurately the usual intake status with a single-day survey through the 24-h recall method. 

In conclusion, adolescents who did not drink milk and had a low level of physical activity were less likely to have a high BMD than those who drank milk and had a high level of physical activity. These results show that there is a synergistic effect of physical activity and milk intake on BMD, suggesting that practicing both moderate to vigorous physical activity and milk consumption in adolescence is an effective way to build healthy bones. The findings of the present study are expected to be useful as empirical data for establishing strategies for promoting healthy bone growth during adolescence.

## Figures and Tables

**Figure 1 nutrients-13-00731-f001:**
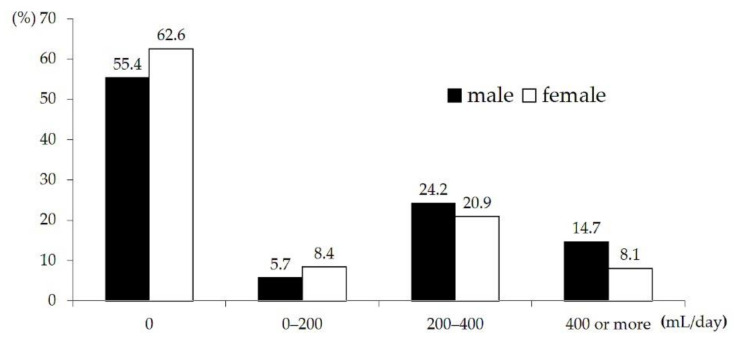
Distribution of daily milk intake.

**Figure 2 nutrients-13-00731-f002:**
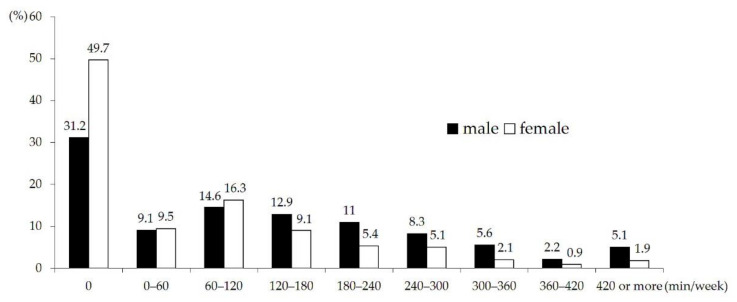
Distribution of physical activity.

**Table 1 nutrients-13-00731-t001:** Sociodemographic characteristics of the subjects.

Variables		M_no_P_low_ ^1^	M_no_P_high_	M_yes_P_low_	M_yes_P_high_	Total	*p*-Value ^2^
School year	Middle school (7th–9th year)	146(34.6) ^3^	100(49.5)	108(41.2)	102(58.3)	456(43.0)	<0.001
	High school (10th–12th year)	276(65.4)	102(50.5)	154(58.8)	73(41.7)	605(57.0)	
Gender	Male	203(48.1)	105(52.0)	130(49.6)	119(68.0)	557(52.5)	0.044
	Female	219(51.9)	97(48.0)	132(50.4)	56(32.0)	504(47.5)	
Income	Low	125(30.0)	49(24.7)	65(25.0)	35(20.2)	274(26.1)	0.086
	Middle–low	108(25.9)	47(23.7)	60(23.1)	43(24.9)	258(24.6)	
	Middle–high	96(23.0)	51(25.8)	61(23.4)	42(24.3)	250(23.9)	
	High	88(21.1)	51(25.8)	74(28.5)	53(30.6)	266(25.4)	
Region (Living area)	Large city	172(40.8)	76(37.6)	105(40.1)	80(45.7)	433(40.8)	0.719
	Medium or small city	183(43.3)	98(48.5)	113(43.1)	77(44.0)	471(44.4)	
	Rural area	67(15.9)	28(13.9)	44(16.8)	18(10.3)	157(14.8)	

^1^ M_no_P_low_: no milk intake + low physical activity; M_no_P_high_: no milk intake + high physical activity; M_yes_P_low_: milk intake + low physical activity; M_yes_P_high_: milk intake + high physical activity (P_low_: physical activity less than 50th percentile; P_high_: physical activity of 50th percentile or more); ^2^
*p*-value by chi-square test. ^3^
*n* (%).

**Table 2 nutrients-13-00731-t002:** Milk intake and physical activity according to the group.

Gender	Variables	M_no_P_low_ ^1^	M_no_P_high_	M_yes_P_low_	M_yes_P_high_	*p*-Value ^2^	Total
Male	*N*	203	105	130	119		557
	Milk intake (mL/day)	0.0 ± 0.0 ^3 a 4^	0.0 ± 0.0 ^a^	360.1 ± 30.3 ^b^	349.0 ± 28.1 ^b^	<0.001	146.2 ± 12.2
	Physical activity (min/week)	11.8 ± 1.9 ^a^	227.3 ± 12.2 ^b^	26.3 ± 3.7 ^c^	230.0 ± 14.3 ^b^	<0.001	120.7 ± 7.9
Female	*N*	219	97	132	56		504
	Milk intake (mL/day)	0.0 ± 0.0 ^a^	0.0 ± 0.0 ^a^	280.8 ± 19.6 ^b^	278.8 ± 18.6 ^b^	<0.001	100.1 ± 9.1
	Physical activity (min/week)	0.0 ± 0.0 ^a^	130.7 ± 11.1 ^b^	0.0 ± 0.0 ^a^	175.9 ± 22.7 ^c^	<0.001	70.4 ± 7.9

^1^ M_no_P_low_: no milk intake + low physical activity; M_no_P_high_: no milk intake + high physical activity; M_yes_P_low_: milk intake + low physical activity; M_yes_P_high_: milk intake + high physical activity (P_low_: physical activity less than 50th percentile; P_high_: physical activity of 50th percentile or more); ^2^
*p*-value by PROC SURVEYREG adjusted for age, body mass index, and energy intake; ^3^ Mean ± SE; ^4 abc^: values with different alphabets in the same row are significantly different at *p* = 0.05 by a Bonferroni test.

**Table 3 nutrients-13-00731-t003:** Physical characteristics of subjects.

Gender	Variables	M_no_P_low_ ^1^	M_no_P_high_	M_yes_P_low_	M_yes_P_high_	*p*-Value ^2^	Total
Male	Age (year)	15.9 ± 0.2 ^3 a 4^	15.6 ± 0.2 ^ab^	15.3 ± 0.2 ^b^	15.1 ± 0.2 ^b^	0.005	15.5 ± 0.1
	Height (cm)	171.0 ± 0.9	172.0 ± 0.8	169.9 ± 0.9	170.4 ± 0.7 ^NS 5^	0.416	171.2 ± 0.4
	Weight (kg)	61.3 ± 1.2	64.1 ± 1.3	62.5 ± 1.7	62.3 ± 1.4 ^NS^	0.745	62.5 ± 0.6
	BMI (kg/m^2^) ^6^	20.9 ± 0.4	21.8 ± 0.4	21.1 ± 0.5	21.4 ± 0.4 ^NS^	0.307	21.2 ± 0.2
	%Fat (%)	20.2 ± 0.7	21.3 ± 0.7	20.0 ± 0.8	22.2 ± 0.9 ^NS^	0.425	20.8 ± 0.4
Female	Age (year)	15.9 ± 0.2 ^a^	15.3 ± 0.2 ^bc^	15.3 ± 0.2 ^bc^	14.8 ± 0.2 ^b^	<0.001	15.4 ± 0.1
	Height (cm)	159.7 ± 0.6	160.4 ± 0.5	160.4 ± 0.7	160.0 ± 0.6 ^NS^	0.551	160.2 ± 0.3
	Weight (kg)	53.0 ± 0.8	56.2 ± 1.1	52.5 ± 1.1	55.3 ± 1.7 ^NS^	0.890	54.1 ± 0.6
	BMI (kg/m^2^)	20.8 ± 0.3 ^a^	21.8 ± 0.3 ^b^	20.3 ± 0.3 ^a^	21.6 ± 0.6 ^b^	0.036	21.0 ± 0.2
	%Fat (%)	31.9 ± 0.5	33.6 ± 0.6	31.8 ± 0.6	33.0 ± 1.0 ^NS^	0.195	32.7 ± 0.4

^1^ M_no_P_low_: no milk intake + low physical activity; M_no_P_high_: no milk intake + high physical activity; M_yes_P_low_: milk intake + low physical activity; M_yes_P_high_: milk intake + high physical activity (P_low_: physical activity less than 50th percentile; P_high_: physical activity of 50th percentile or more); ^2^
*p*-value by PROC SURVEYREG adjusted for age, body mass index (BMI) and energy intake; ^3^ Mean ± SE; ^4 abc^: values with different alphabets in the same row are significantly different at *p* = 0.05 by Bonferroni test; ^5 NS^: not significant; ^6^ BMI = weight(kg)/height(m^2^).

**Table 4 nutrients-13-00731-t004:** Relationships of bone mineral density with milk intake and physical activity.

Milk Intake and Physical Activity	Variables of BMD ^1^	Male (*n* = 557)		Female (*n* = 504)	
		r	*p*-Value ^2^	r	*p*-Value
Milk intake	Total BMD (g/cm^2^)	0.025	0.574	0.038	0.426
	Femur BMD (g/cm^2^)	0.017	0.707	0.017	0.718
	Femur neck BMD (g/cm^2^)	−0.001	0.991	0.029	0.545
	Lumbar BMD (g/cm^2^)	0.049	0.274	0.075	0.112
Physical activity time	Total BMD (g/cm^2^)	0.212	<0.001	0.120	0.019
	Femur BMD (g/cm^2^)	0.257	<0.001	0.142	0.005
	Femur neck BMD (g/cm^2^)	0.250	<0.001	0.135	0.008
	Lumbar BMD (g/cm^2^)	0.120	0.020	0.180	<0.001

^1^ BMD: bone mineral density; ^2^
*p*-value by partial correlation controlled by age, body mass index, and energy intake.

**Table 5 nutrients-13-00731-t005:** Bone mineral density among the groups of the combination of milk intake and physical activity.

Gender	Variables	M_no_P_low_ ^2^	M_no_P_high_	M_yes_P_low_	M_yes_P_high_	*p*-Value ^3^	Total
Male	Total BMD ^1^ (g/cm^2^)	0.916 ± 0.012 ^4 a 5^	0.952 ± 0.010 ^b^	0.917 ± 0.014 ^ac^	0.947 ± 0.012 ^bc^	0.003	0.939 ± 0.005
	Femur BMD (g/cm^2^)	0.896 ± 0.015 ^a^	0.952 ± 0.016 ^b^	0.917 ± 0.017 ^ab^	0.952 ± 0.013 ^b^	0.003	0.934 ± 0.006
	Femur neck BMD (g/cm^2^)	0.813 ± 0.014 ^a^	0.866 ± 0.014 ^b^	0.817 ± 0.018 ^ac^	0.863 ± 0.014 ^c^	0.002	0.847 ± 0.006
	Lumbar BMD (g/cm^2^)	0.839 ± 0.015 ^a^	0.881 ± 0.016 ^b^	0.850 ± 0.020 ^ab^	0.866 ± 0.015 ^b^	0.019	0.865 ± 0.007
Female	Total BMD (g/cm^2^)	0.868 ± 0.010	0.866 ± 0.009	0.860 ± 0.010	0.866 ± 0.011 ^NS 6^	0.094	0.866 ± 0.005
	Femur BMD (g/cm^2^)	0.868 ± 0.014	0.876 ± 0.012	0.859 ± 0.015	0.885 ± 0.011 ^NS^	0.416	0.873 ± 0.006
	Femur neck BMD (g/cm^2^)	0.760 ± 0.013	0.769 ± 0.014	0.755 ± 0.015	0.770 ± 0.011 ^NS^	0.700	0.765 ± 0.006
	Lumbar BMD (g/cm^2^)	0.902 ± 0.011 ^a^	0.900 ± 0.013 ^a^	0.898 ± 0.014 ^a^	0.931 ± 0.015 ^b^	0.030	0.904 ± 0.007

^1^ BMD: bone mineral density; ^2^ M_no_P_low_: no milk intake + low physical activity; M_no_P_high_: no milk intake + high physical activity; M_yes_P_low_: milk intake + low physical activity; M_yes_P_high_: milk intake + high physical activity (P_low_: physical activity less than 50th percentile, P_high_: physical activity of 50th percentile or more); ^3^
*p*-value by PROC SURVEYREG adjusted for age, body mass index, and energy intake; ^4^ Mean ± SE; ^5 abc^: Values with different alphabets in the same row are significantly different at *p* = 0.05 by Bonferroni test; ^6 NS^: not significant.

**Table 6 nutrients-13-00731-t006:** Odds ratios on the bone mineral density according to the combination of milk intake and physical activity.

Variables	Group	Male		Female	
		Odds Ratio	CI ^1^	Odds Ratio	CI
Total BMD ^2^ ≥50th percentile, 0.896 (Reference)	M_no_P_low_ ^3^ vs. M_yes_P_high_	0.317	(0.151, 0.663) * ^4^	0.635	(0.301, 1.342)
	M_no_P_high_ vs. M_yes_P_high_	1.539	(0.729, 3.249)	0.654	(0.332, 1.289)
	M_yes_P_low_ vs. M_yes_P_high_	0.582	(0.248, 1.363)	0.652	(0.307, 1.384)
Femur BMD ≥50th percentile, 0.897 (Reference)	M_no_P_low_ vs. M_yes_P_high_	0.289	(0.154, 0.539) *	0.684	(0.339, 1.380)
	M_no_P_high_ vs. M_yes_P_high_	0.773	(0.394, 1.517)	0.752	(0.388, 1.461)
	M_yes_P_low_ vs. M_yes_P_high_	0.485	(0.234, 1.007)	0.545	(0.261, 1.141)
Femur neck BMD ≥50th percentile, 0.801 (Reference)	M_no_P_low_ vs. M_yes_P_high_	0.512	(0.274, 0.958) *	0.843	(0.405, 1.754)
	M_no_P_high_ vs. M_yes_P_high_	1.551	(0.798, 3.012)	0.918	(0.454, 1.857)
	M_yes_P_low_ vs. M_yes_P_high_	0.636	(0.30, 1.35)	0.894	(0.409, 1.951)
Lumbar BMD ≥50th percentile, 0.875 (Reference)	M_no_P_low_ vs. M_yes_P_high_	0.493	(0.245, 0.992) ^*^	0.433	(0.21, 0.895) ^*^
	M_no_P_high_ vs. M_yes_P_high_	1.140	(0.58, 2.24)	0.485	(0.233, 1.009)
	M_yes_P_low_ vs. M_yes_P_high_	1.149	(0.568, 2.322)	0.434	(0.203, 0.928) ^*^
	M_yes_P_high_	1.000	(ref)	1.000	(ref)

^1^ CI: confidence interval; ^2^ BMD: bone mineral density; ^3^ M_no_P_low_: no milk intake + low physical activity; M_no_P_high_: no milk intake + high physical activity; M_yes_P_low_: milk intake + low physical activity; M_yes_P_high_: milk intake + high physical activity (P_low_: physical activity less than the 50th percentile; P_high_: physical activity of the 50th percentile or more); ^4^ *: *p*<0.05 by PROC SURVEYLOGISTIC.

## Data Availability

Data were obtained from the Korean National Health and Nutrition Examination Survey (KNHANES) and are available from the KNHANES website (at http://knhanes.cdc.go.kr (accessed on 26 June 2017)).

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
