# Peer review of "The Combined Effects of Milk Intake and Physical Activity on Bone Mineral Density in Korean Adolescents"

_nutrients, 2021, doi:10.3390/nu13030731_

Round 1
Reviewer 1 Report
Overall this is a well-written paper with an interesting result in the nutrition area.
INTRODUCTION
The introduction provides sufficient background information for readers to understand the problem, however, the authors should clarify the importance of physical activity and milk in this age. This part would improve this section.
Motivations for this study are more than clear. The objectives are clearly defined at the Introduction and the argumentation in this last part was concise and clarifying. It would be recommended to include a study hypothesis.
METHODS
The experimental approach is appropriate for the aim of the study.
This section is well described and allows to replicate the study.
RESULTS
Results paragraphs include more relevant and extended data.
All the tables include specific, well-developed statistic.
DISCUSSION
All possible interpretations of the data considered are consistent.
explain limitation, vitamin D control, physical activity recording, nutritional recording.
Include practical application section
Provide a concise conclusion responding study aim
LITERATURE CITED
The literature cited is relevant to the study.
SIGNIFICANCE AND NOVELTY
As it stands, the results are novel and important enough for this journal.
Author Response
Overall this is a well-written paper with an interesting result in the nutrition area.
-- Thank you for taking the time to review this manuscript and thank you for the meaningful review. The parts you pointed out for revision have been modified as much as possible and marked in red in the text.
INTRODUCTION
The introduction provides sufficient background information for readers to understand the problem, however, the authors should clarify the importance of physical activity and milk in this age. This part would improve this section.
-- Considering your suggestion, we corrected later part of the introduction section as follows.
'As described above, various studies have been conducted on the effects of dietary intake or physical activity on BMD during the growth period. On the other hand, few studies have examined the combined effects of milk intake and physical activity. Limited trial studies suggested an important interaction between physical activity and the intake of dietary calcium to increase bone mass. When exercise and milk intake were accompanied, bone density formation was greater than either exercise or calcium intake alone.'
Motivations for this study are more than clear. The objectives are clearly defined at the Introduction and the argumentation in this last part was concise and clarifying. It would be recommended to include a study hypothesis.
-- We added this sentence to the introduction section.
' We hypothesized that adolescents who were active while ingesting milk would have higher BMD than adolescents who practiced only one of these or not all of them.'
METHODS
The experimental approach is appropriate for the aim of the study.
-- Thank you.
This section is well described and allows to replicate the study.
-- Thank you..
RESULTS
Results paragraphs include more relevant and extended data.-
-- Thanks for the very good comments. The contents of the results were revised more concisely as follows.;
Table 3 lists the general characteristics of each of the four groups classified according to milk intake and the level of physical activity. In both males and females, the mean age was highest in the MnoPlow group and lowest in the MyesPhigh. For body mass index (BMI), there was a significant difference only in females, showing that the MnoPlow and MyesPlow groups with low levels of physical activity had a significantly lower mean BMI than the groups with high levels of physical activity. There was no difference in body fat (%) between groups in both male and females.
All the tables include specific, well-developed statistic.
-- Thank you..
DISCUSSION
All possible interpretations of the data considered are consistent.
-- Thank you
explain limitation, vitamin D control, physical activity recording, nutritional recording.
--Thank you for making an important point. Those points were added to the discussion section and marked in red.
Provide a concise conclusion responding study aim
-- We concluded as follows.
'In conclusion, adolescents who did not drink milk and had a low level of physical activity were less likely to have a high BMD than those who drank milk and had a high level of physical activity. These results show that there is a synergistic effect of physical activity and milk intake on BMD, suggesting that practicing both moderate to vigorous physical activity and milk consumption in adolescence is an effective way to build healthy bones. The findings of the present study are expected to be useful as empirical data for establishing strategies for promoting healthy bone growth during adolescence.'
LITERATURE CITED
The literature cited is relevant to the study.
--- Thank you
SIGNIFICANCE AND NOVELTY
As it stands, the results are novel and important enough for this journal.- Thank you for your very meaningful review.

Reviewer 2 Report
This is a well written article and an interesting research that presents some limitations that should be clarified.
Introduction
The introduction is well written and clearly presents the topic.
Line 57. I would be cautious suggesting that milk consumption is the most commonly used intervention methods for healthy bone growth. I would say dietetic interventions, not just milk consumption.
Methods
Line 117. Rcomputing?
Is the KNHANES questionnaire validated? If not please add as a limitation. For example I would say volleyball and badminton are vigorous physical activities, while moving with heavy objects could be classified as moderate.
2.4 Regarding the subject grouping classification, it is a bit arbitrary to use de median to classify the physical activity groups. Could you please test what happens when you use compliance with physical activity guidelines? That is children meeting 60 minutes of moderate or vigorous activity.
2.5 BONE MINERAL DENSITY. Please specify the performed scans. It is clear you performed a whole body scan, but did you also perform a hip scan? Or are the hip values obtained from the whole body scan?
2.6 Statistical analysis. I´m not familiarized with the The proc survey multiple logistic regression analysis and therefore cannot assess if it is adequate for your data.
If possible it would be interesting to control for Vitamin D, and calcium levels. What I mean is that you can obtain calcium from other sourcers, not just milk (e.g. nuts, cheese….). Therefore it is possible that a boy/girls doesn´t like milk but does attain the recommended calcium and vitamin D levels through other sources. It would be interesting to control that or at least to see if it is influencing the results in any way.
Discussion. Line 320. The statement you perform (BMD and activities with and without impacts) is correct but you are missing a reference or two. There are several reviews and meta-analyses on the topic.
Line 323. Again you cite old studies, when there are updated reviews on this topic. Please update your citations.
Line 333. If you collected vigorous and moderate physical activity separately. Would it not have made sense to also analyze them independently. If this was done it is possible that the correlation for vigorous PA would be high, and that for moderate would be very low or non significant. Maybe that is why you are obtaining low correlations.
Line 344. Again you cite two studies to talk about vegetarians and bones. In this case both are meta-analysis, but none talk about vegetarians, please include a study with vegetarians and bone in which the find the things that you are stating.
Limitations. Another important limitation is that you not take into account the type of activity. For example swimming or cycling can be a moderate or vigorous activities, but have a little or no effect on bone mass. This could bias your results. Please comment on the limitation section.
You only perform on 24-hour recall. Literature suggests a minimum of 3. Please state as a limitation.
Author Response
This is a well written article and an interesting research that presents some limitations that should be clarified.
-- Thank you for taking the time to review this manuscript and thank you for the meaningful review. The parts you pointed out for revision have been modified as much as possible and marked in blue in the text.
INTRODUCTION
The introduction is well written and clearly presents the topic.
--Thank you
Line 57. I would be cautious suggesting that milk consumption is the most commonly used intervention methods for healthy bone growth. I would say dietetic interventions, not just milk consumption.
-- We corrected as it bellows
"Among these factors, milk or calcium consumption and physical activity are important factors for healthy bone growth"
METHODS
Line 117. Rcomputing?
-- It was typing error and also unnecessary word, so we erased it
Is the KNHANES questionnaire validated? If not please add as a limitation. For example I would say volleyball and badminton are vigorous physical activities, while moving with heavy objects could be classified as moderate.
-- The KNHANES is based on the IPAQ short form and the classification of physical activity intensity of American College of Sports Medicine (ACSM). The Korean version of IPAQ was accepted as a proper one by the International Physical Activity Questionnaire (IPAQ) developers (Oh et al., 2007). According to ACSM (American College of Sports Medicine)'s classification of physical activity by intensity, moderate-intensity activity corresponds to 3.0-6.0 METs, and badminton (recreational, 4.5 METs), slow swimming (6.0 METs), tennis doubles (5.0 METs), vollyball (noncompetitive, 3.0- 4.0 METs) included. Vigorous activity is an activity that exceeds 6 METs, so carrying heavy loads (7.5 METs) falls into this category.
2.4 Regarding the subject grouping classification, it is a bit arbitrary to use de median to classify the physical activity groups. Could you please test what happens when you use compliance with physical activity guidelines? That is children meeting 60 minutes of moderate or vigorous activity
-- Thanks for the very good comments. We also thought about classifying exercise groups in the same way you pointed out. However, 60 minutes or more of moderate to vigorous intensity physical activity per day is equivalent to 420 minutes per week. In subjects of this study, as shown in Figure 2, the adolescents who performed physical activity for more than 420 minutes per week were 5.1% in male and 1.9% in female, most of which did not reach the guidelines. if the groups are divided based on the recommended level of physical activity, the difference in the number of people between the four groups becomes remarkable, making statistical analysis impossible. Therefore, in this study, subjects were grouped based on exercise time rather than exercise intensity.
2.5 BONE MINERAL DENSITY. Please specify the performed scans. It is clear you performed a whole body scan, but did you also perform a hip scan? Or are the hip values obtained from the whole body scan?
-- Thanks for the very good comments. The following bone density measurement method was added to the manuscript methodology.
"BMD was measured using dual-energy X-ray absorptiometry (DXA; DISCOVERY-W fan-beam densitometer Hologic Inc., Bedford, MA, USA) and whole body, lumbar spine, and femur were scanned. When measuring the lumbar spine, a lumbar positioner was used to reduce spinal lordosis, and the lumbar spine was positioned straight so as to be in line with the vertical central axis of the image. The image included the mid-section of T12 and L5, and to determine whether the lumbar spine was correctly positioned, it was checked whether the 12th rib and iliac crest were visible in the image, and whether the intervertebral disc of L4-L5 passed in line with the iliac crest. When measuring the femur, the angle of the leg was adjusted so that the femoral shaft was positioned straight in line with the vertical central axis of the image. When measuring DXA, it was checked if there were any artifacts such as coins or keys in the pocket, such as buttons, wires, jewelry, metal objects in the pocket. Among the various DXA measurement indices, total body, femur, femur neck, and lumbar spine (L1-4) BMD were analyzed statistically and total body BMD was calculated using the BMD values of the whole body except for head BMD".
2.6 Statistical analysis. I´m not familiarized with the The proc survey multiple logistic regression analysis and therefore cannot assess if it is adequate for your data.
-- Thankou for your comment. The sampling method of the National Health and Nutrition Survey is not simple random sampling, but is sampling by stratified, clustered, and systematic sampling. Correct conclusions can be drawn only when statistical analysis is conducted in consideration of the extraction rate, response rate, and weight reflecting the population structure. Therefore, it was analyzed using complex sample design procedures of SAS such as Proc Surveyfreq, Proc Survemean, Proc Surveyreg, and Proc Surveylogistic instead of ANOVA, chi-square test, regression or logistic test using simple random sampling in SAS
If possible it would be interesting to control for Vitamin D, and calcium levels. What I mean is that you can obtain calcium from other sourcers, not just milk (e.g. nuts, cheese….). Therefore it is possible that a boy/girls doesn´t like milk but does attain the recommended calcium and vitamin D levels through other sources. It would be interesting to control that or at least to see if it is influencing the results in any way.-
-- Thank you for pointing out a very important thing. The below content was added to the discussion section.
"It was confirmed that the calcium intake of Korean adolescents was very low, but milk was the main source of calcium. In addition, when the bone mineral density variable was analyzed using calcium as a parameter, the same result as milk intake was obtained. Thus milk intake instead of calcium intake was presented in this paper. In addition, vitamin D intake was excluded from the control variable as there was no significant difference between groups."
discussion sectionã…£
Discussion. Line 320. The statement you perform (BMD and activities with and without impacts) is correct but you are missing a reference or two. There are several reviews and meta-analyses on the topic.
-- Thank you. The following references related to this topic were added.
Ahmed Elhakeem, Jon Heron, Jon H. Tobias, D. A. L. Physical Activity throughout Adolescence and Peak Hip Strength in Young Adults. JAMA Netw. 2020, 3 (8), e2013463.
Gomez-Bruton, A.; Montero-Marín, J.; González-Agüero, A.; García-Campayo, J.; Moreno, L. A.; Casajús, J. A.; Vicente-Rodríguez, G. The Effect of Swimming During Childhood and Adolescence on Bone Mineral Density: A Systematic Review and Meta-Analysis. Sport. Med. 2016, 46 (3), 365–379. https://doi.org/10.1007/s40279-015-0427-3.
Line 323. Again you cite old studies, when there are updated reviews on this topic. Please update your citations.
-- Thankyou. We put the following references.
Weaver, C. M.; Gordon, C. M.; Janz, K. F.; Kalkwarf, H. J.; Lappe, J. M.; Lewis, R.; O’Karma, M.; Wallace, T. C.; Zemel, B. S. The National Osteoporosis Foundation’s Position Statement on Peak Bone Mass Development and Lifestyle Factors: A Systematic Review and Implementation Recommendations. Osteoporos. Int. 2016, 27 (4), 1281–1386. https://doi.org/10.1007/s00198-015-3440-3.
American College of Sports Medicine, ACSM’s Guidelines for Exercise Testing and Prescription, 10th ed.; Lippincott: Philadel-phia, PA, 2017.
Line 333. If you collected vigorous and moderate physical activity separately. Would it not have made sense to also analyze them independently. If this was done it is possible that the correlation for vigorous PA would be high, and that for moderate would be very low or non significant. Maybe that is why you are obtaining low correlations.
-- Thanks for the very good comments.
Perhaps, as you pointed out, analyzing the correlation between the intensity of exercise and bone density is also very meaningful. However, as mentioned above, the study subjects were grouped by exercise time (combined medium and high-intensity exercise time) rather than exercise intensity, so the correlation between exercise time and bone density was investigated according to the purpose of the study. In Table 4, physical activity was modified to physical activity time.
Line 344. Again you cite two studies to talk about vegetarians and bones. In this case both are meta-analysis, but none talk about vegetarians, please include a study with vegetarians and bone in which the find the things that you are stating.
-- Thank you. We deleted the phrase 'such as vegetarians' because we found it would be controversial to refer to vegetarian as an example of insufficient intake of protein and minerals after investigating the dietary intake of vegetarian.
Limitations. Another important limitation is that you not take into account the type of activity. For example swimming or cycling can be a moderate or vigorous activities, but have a little or no effect on bone mass. This could bias your results. Please comment on the limitation section.
-- We added the following phrases to the limitation.
'"The KNHANES used in this study does not investigate type of physical activity. Therefore, non-weight bearing physical activity participation time, such as 'slow swimming' included as an example of moderate intensity activity, could not be considered separately to be analyzed, possibly reducing the correlation between physical activity and BMD.'"
You only perform on 24-hour recall. Literature suggests a minimum of 3. Please state as a limitation.
-- We added the following phrases to the limitation.
"This study has the limitation of using a single day of 24-hour recall. Unfortunately, at that time, the KNHNES provided an only a single day of 24-hour recall data. One of the important drawbacks of the 24-hour recall method is that it is very difficult to grasp an individual's usual intake status with 24-hour recall data with a single day due to dietary variation within individuals..--- This content was added to the discussion section"
